# Impact of temperature on Downs herring (*Clupea harengus*) embryonic stages: First insights from an experimental approach

**Lola Toomey** *, **Carolina Giraldo**, **Christophe Loots**, **Kélig Mahé**, **Paul Marchal**, **Kirsteen MacKenzie**

IFREMER, Channel and North Sea Fisheries Research Unit, Boulogne-sur-Mer, France

* Lola.Toomey@ifremer.fr

## Abstract

Among all human-induced pressures, ocean warming is expected to be one of the major drivers of change in marine ecosystems. Fish species are particularly vulnerable during embryogenesis. Here, the impact of temperature was assessed on embryonic stages of Atlantic herring (*Clupea harengus*), a species of high socio-economic interest, with a particular focus on the under-studied eastern English Channel winter-spawning component (Downs herring). Key traits linked to growth and development were experimentally evaluated at three temperatures (8°C, 10°C and 14°C), from fertilization to hatching, in standardized controlled conditions. Overall negative impacts of increased temperature were observed on fertilization rate, mean egg diameter at eyed stage, hatching rate and yolk sac volume. A faster developmental rate and a change in development stage frequency of newly hatched larvae were also observed at higher temperature. Potential parental effects were detected for four key traits (i.e. fertilization rate, eyed survival rate, mean egg diameter and hatching rate), despite a limited number of families. For instance, a large variability among families was shown in survival rate at eyed stage (between 0 and 63%). Potential relationships between maternal characteristics and embryo traits were therefore explored. We show that a substantial proportion of variance (between 31 and 70%) could be explained by the female attributes considered. More particularly, age, traits linked to life history (i.e. asymptotic average length and Brody growth rate coefficient), condition and length were important predictors of embryonic key traits. Overall, this study constitutes a stepping-stone to investigate potential consequences of warming on Downs herring recruitment and provides first insights on potential parental effects.

## 1. Introduction

Since the industrial revolution, greenhouse gas emissions have drastically altered the oceans [1, 2]. The sixth assessment report of the Intergovernmental Panel on Climate Change (IPCC) shows that human-induce changes are larger than previously expected [2]. Among the human-induced changes [2–4], ocean warming caused by carbon dioxide emissions is a major driver

**Funding:** This work was financially supported by the European Union (ERDF), the French State, the French Region Hauts-de-France and Ifremer, in the framework of the project CPER MARCO 2015-2021. The funders had no role in study design, data collection and analysis, decision to publish, or preparation of the manuscript.

**Competing interests:** The authors have declared that no competing interests exist.

**Abbreviations:** DNA, Deoxyribonucleic acid; EEC, Eastern English Channel; ELS, Early-Life Stages; FC, Fulton condition index; GSI, Gonadosomatic index; HSI, Hepatosomatic index; IPCC, Intergovernmental Panel on Climate Change; K, Brody growth rate coefficient; OE, Oocyte ellipticity; RNA, Ribonucleic acid; SL, Standard Length; TL, Total length; TL∞, Asymptotic average length; W, Wet weight; YSV, Yolk sac volume.

of change in marine ecosystems [4–8], with sea surface temperatures increasing by 0.25–1˚C on average (depending on location), and consequences are already visible at all levels of biological organization [2, 8, 9–11]. Fish, which are poikilothermic, are particularly affected by seawater temperature changes as their metabolism relies on temperature-dependent reactions [12, 13]. Changes in physiology and behavior, for instance, have been reported, with consequences for spatial distribution, abundance (i.e. mortality), reproductive ability, and growth [14–19]. These changes due to climate change stressors (e.g. ocean warming) can impact the fisheries sector, with potential economic consequences, and threaten human food security [15, 20]. Evaluating species' vulnerability is therefore crucial to implement appropriate management measures and develop tools to mitigate negative consequences.

Assessing the response of species to ocean warming is a major and complex challenge. In situ studies can allow for long-term monitoring of populations [21], but the marine environment is complex and the effects of different environmental factors on individual phenotypes are confounded [22]. Experiments under controlled conditions are therefore essential to disentangle effects and study interactions between drivers [23, 24], to improve understanding of phenomena observed in situ and potentially improve predictive models, while providing a stepping-stone to further work at expanded ecological levels.

Species' vulnerability to global warming is dependent upon the most temperature-sensitive stages in the life cycle. Embryos are sensitive life stages due to lower thermal tolerances, partly linked to ontogenetic changes in aerobic capacity [25]. Early-life stages (ELS), which include both embryos and larvae, are particularly vulnerable to warming as behavioral responses are limited (no or limited swimming ability [8]). ELS mortalities also strongly determine recruitment variability [26, 27], the understanding of which is essential for fisheries management [28]. Finally, conditions experienced during embryogenesis can also have consequences on later-stage performances [29, 30]. The embryonic period has therefore been the focus of many studies for marine species of fisheries interest [31].

Among the species of interest, Atlantic herring (*Clupea harengus*) is a socio-economically important pelagic fish species, with a wide geographic distribution and various life-history strategies depending on the surrounding environment [32, 33]. Atlantic herring usually reach maturity between two and three years old and are considered a single-batch spawner with a demersal egg phase [32, 34]. These herring have great plasticity in reproduction strategies and exhibit reproductive homing behavior [34, 35]. In the North Sea, four spawning components have been described based on their reproductive timing and spawning grounds: Shetland/Orkney, Buchan, and Banks, which all spawn largely in autumn, and Downs, which largely spawn in winter [34, 36, 37]. These components share the same feeding grounds outside of their reproductive seasons [33, 36, 37], and are thus managed as a single stock unit [34].

Responses to different climate change drivers are particularly well-studied in several stock components of Atlantic herring [31]. Studies on ELS have mainly focused on the impacts of temperature and salinity, often individually (e.g. [38–44]). Variable impacts of temperature rise have been experimentally shown for various traits linked to different biological functions. For instance, higher temperature accelerated development, impaired hatching rate and resulted in smaller larvae at hatching in Norwegian populations [39, 41]. With increased temperature, smaller larvae also emerged at hatching for Baltic herring, but the impact on hatching rate was more limited when the same temperature range was considered [44]. Overall, the response to temperature varies across studies and stock components. Most of the studies investigating the impact of temperature on ELS have focused on Norwegian, Baltic or Western Scotland stock units, with very few studies focused on North Sea components. Differential responses may, however, be expected at the intraspecific scale and lead to divergent recommendations for conservation and management of wild components [7, 45]. Indeed, inter-

population differentiation is common across fish species, whether marine (e.g. [46, 47]) or freshwater species (e.g. [48, 49]). In addition to differentiation seen across life stages (inter-individual variation), intraspecific variation in temperature tolerance can originate from phenotypic plasticity (i.e. the ability of a genotype to produce different phenotypes in response to environmental changes) and/or heritable genetic differences [45]. Different responses to warming could therefore be seen across the different herring spawning components/stock units.

Parental effects may also affect the response to stressors, as they influence various early-life history traits and may have a substantial impact on recruitment [50]. Although paternal effects have recently received more attention and are nonnegligible [50, 51], maternal effects have been widely investigated and appear to be particularly important and often predominant [39, 52–55]. These effects are usually defined as non-genetic contribution to offspring and contribute greatly to population phenotypic variation [52]. They are attributed to numerous characteristics (e.g. age, condition or length), that trigger variability in embryonic and larval quality [39]. Finding potential maternal predictors of offspring quality could contribute to the estimation of fish larvae production and recruitment success [56].

Downs herring are relatively understudied in comparison to other stock components (but see inter-stock units study in [57, 58]); unlike the other North Sea components, they spawn in winter in the eastern English Channel (EEC) and southern part of the North Sea. These herring have different life-history strategies and ecological niches in comparison to other well-studied stock components, such as Norwegian or Baltic herring, and require further component-specific study. The Downs component is poorly considered in recruitment evaluation, despite being a major contributor to the North Sea stock [59, 60]. Further information on Downs herring is thus essential to assess and manage the North Sea stock adequately. A preliminary evaluation of the response to climate change (warming and acidification) was previously conducted on Downs larval stages [61]. Here, the scope of that study was expanded by conducting assessment on the sensitive embryonic period and investigating potential parental effects on embryonic traits. We aimed to: (i) evaluate the effect of different water temperatures across major stages during embryogenesis, and (ii) provide first insights on parental effects and evaluate the relationships between maternal characteristics and embryo traits. To achieve these goals, embryonic development was tracked, from fertilization to hatching, across three temperature regimes under standardized controlled environmental conditions.

## 2. Materials and methods

### 2.1. Temperature scenarios

Three temperature scenarios were tested in this study: 8˚C, 10˚C and 14˚C. The first two temperatures are within the preference range of Atlantic herring larvae [37, 44]. These two scenarios represent current conditions encountered by wild embryos during their incubation and hatching period (mid-November 2021-January 2022; S1 Fig; [59, 62]), while the International Council for the Exploration of the Sea data (ICES) estimated the average 1992–2020 temperature in the EEC at 9.35 ± 2.20˚C (ICES Data portal, accessed the 4th of May 2022) during the herring spawning period. Our first two temperature scenarios are therefore representative of temperatures experienced during the spawning season and constitute control scenarios. The 14˚C regime corresponds to a future scenario following IPCC predictions and more precisely considering the SSP5-8.5 prediction (Shared Socioeconomic Pathway—fossil-fueled development [2]). This scenario forecasts a 3˚C increase which, considering the upper value of the EEC average temperature range (11.55˚C), led us to test 14˚C.

## 2.2. Collection of biological material & the experimental phase

The experiment was performed at the Nausicaá aquarium in Boulogne-sur-Mer between December 2021 and January 2022. Wild, ready-to-spawn herring from the EEC were obtained from local fishers operating pelagic trawls. Herring were collected during the last haul before returning to harbor. Downs herring were collected at two different dates: 8th of December 2021 (first artificial fertilization experiment/first batch) and 17th of December 2021 (second artificial fertilization experiment/second batch). The two batches overlapped in experiment time. Ripe dead herring were transported on ice to the experimental facility. All individuals were measured (total length and standard length, ± sd, 0.1 cm) and weighed (wet weight, ± 0.1 g). Nine females (individually numbered as: F1 to F9; mean standard length: 26.3 ± 0.9 cm, mean wet weight: 228.5 ± 20.9 g) and 27 males (mean standard length: 26.1 ± 1.2 cm, mean wet weight: 219.8 ± 37.3 g) were collected for the first batch. Eleven females (individual numbers: F10 to F20; mean standard length: 24.8 ± 1.5 cm, mean wet weight: 180.6 ± 35.3 g) and 33 males (mean standard length: 22.7 ± 1.8 cm, mean wet weight: 165.8 ± 36.9 g) were used for the second batch.

Artificial fertilization was carried out following a protocol adapted from Joly et al. [63]. Eggs from the different females (n = 20) were stripped separately in seawater at 10˚C and spread on 12 plastic plates (8 cm x 3.2 cm; mean 375 ± 150 eggs per plate; 240 plates considering all females). In order to avoid oxygenation issues during development, we deposited the eggs in single layers as much as possible. One plate per female was randomly chosen for imaging with a binocular microscope (Olympus® SZX16, at 20–40× magnification) for later estimation of mean oocyte diameter. In order to minimize male effects and optimize fertilization success (i.e. lower risks to use bad-quality males), three randomly chosen males were used to fertilize the eggs of each female. For each female, a different pool of males was used. Fertilization was performed by dissecting, weighing and incising male gonads, then spreading the milt across each egg plate out of the water by gently putting in contact the incised gonad and sticky eggs. Egg plates from each family (i.e. parental combination relatively to the different females (F1 to F20); three males x one female for each family) were then placed for ten minutes in three separate trays with seawater at 8, 10 or 14˚C (four plates per temperature scenario and per female). After this incubation, plates were rinsed with seawater to remove non-adherent eggs and rapidly transferred into experimental systems at the same temperature as the incubation (S2 Fig; see 2.3). Females were dissected to remove the remaining gonad and weighed again to estimate the gonad weight. The liver and a piece of muscle taken from behind the dorsal fin were also sampled for all females to estimate the HepatoSomatic Index (HSI) and RNA: DNA ratios, respectively. Finally, sagittal otoliths (from left and right sides) were extracted from the cranial cavity of all fish and cleaned before storage in Eppendorf tubes for subsequent age estimation (see 2.4 for details on all functional traits assessments).

## 2.3. Experimental phase

The study was carried out in accordance with the recommendations in the Guide for the Care and Use of Laboratory Animals of the National Institutes of Health. All procedures followed international and national guidelines for protection of animal welfare (Directive 2010/63/EU). All embryos were killed using an overdose of MS-222 (about 300 mg.L$^{-1}$; endpoint: absence of heartbeat) and all efforts were made to minimize suffering.

Three identical aquatic partial reuse rearing systems were used to test three temperature regimes (S2 Fig). Each system received mechanically filtered (using inline cartridge filters of 10 μm, 5 μm and 1 μm) and UV-treated (Aqua Medic® Helix Max 2.3 36W UV Sterilizer, France) natural seawater from the Nausicaá cold seawater supply. Seawater was discharged into

**Table 1. Daily physical and chemical parameters followed along the experiment for each temperature regime.** Sd: standard deviation.

| Temperature regime | Parameter | Mean | Sd | Inter-incubator sd |
|---|---|---|---|---|
| 8°C | Temperature (°C) | 8.08 | 0.25 | 0.01 |
| | pH | 7.89 | 0.12 | 0.03 |
| | Salinity | 28.57 | 0.70 | 0.01 |
| | Oxygen (mg/L) | 10.79 | 0.56 | 0.05 |
| | Oxygen saturation (%) | 109.84 | 4.11 | 0.40 |
| 10°C | Temperature (°C) | 10.58 | 0.31 | 0.02 |
| | pH | 7.93 | 0.05 | 0.01 |
| | Salinity | 28.54 | 0.67 | 0.03 |
| | Oxygen (mg/L) | 9.93 | 0.48 | 0.04 |
| | Oxygen saturation (%) | 107.07 | 3.98 | 0.90 |
| 14°C | Temperature (°C) | 14.16 | 0.12 | 0.02 |
| | pH (n = 34) | 7.91 | 0.06 | 0.01 |
| | Salinity | 28.62 | 0.34 | 0.05 |
| | Oxygen (mg.L$^{-1}$) | 8.90 | 0.29 | 0.08 |
| | Oxygen saturation (%) | 102.48 | 2.18 | 1.18 |

a 130 L aerated sump tank with a renewal rate of 4.28 L.min$^{-1}$ (S2 Fig). From the sump tank, seawater was pumped in parallel to two 39.35 L black plastic incubators at a rate of 40 L.h$^{-1}$, then returned to the sump by gravity (S2 Fig). Each incubator contained nine 1.86 L hatching cells, where the egg plates from different females were randomly placed (S2 Fig). Each aquatic rearing system was operated at a different temperature, i.e. 8°C, 10°C and 14°C. For the 8°C treatment, a TECO® TK-2000 aquarium chiller was used for temperature control. The 10°C treatment was temperature-controlled by the established water renewal rate of 4.28 L.min$^{-1}$. Finally, for the 14°C treatment, water was heated using two Aqua Medic® 300 W Titanium heaters. Plastic plates containing eggs were randomly spread across hatching cells, incubators and temperature regimes (12 plates in total per female, four plates per temperature scenario and per female) and placed at the bottom of each hatching cell. Photoperiod was 12h/12h L:D (light/dark cycle) and light intensity was 20–30 lux across all treatments. Temperature, pH, salinity, dissolved oxygen and oxygen saturation were measured daily in each incubator (Table 1) using a YSI® professional plus multiparameter instrument. The experiment was ended for each temperature regime when all larvae had hatched and that remaining embryos that did not hatch were dead.

### 2.4. Functional traits assessment

**2.4.1. Maternal traits.** Measurements of total length (TL; cm), standard length (SL; cm) and wet weight (W; g) were taken for all breeders (20 females and 60 males). Fulton's condition factor (CF), reflecting the individual nutritional condition (i.e. recent food availability), was calculated for each fish using the following formula:

$$CF = 100 * \frac{W}{TL^3} \qquad (1)$$

Where W is the wet weight and TL the total length.

The GonadoSomatic Index (GSI), indicative of the reproductive maturity, was calculated for all fish using the following formula:

$$GSI = 100 * \frac{W_G}{W} \qquad (2)$$

Where $W_G$ is the weight of the gonad (g, ± 0.1 g) and W the total wet weight of the fish. For females, $W_G$ was estimated by calculating the difference between total wet weight and wet weight after strip spawning and removing the rest of the gonad.

The HepatoSomatic Index (HSI), indicating energy reserves of individuals, was estimated for females using the following formula:

$$HSI = 100 * \frac{W_L}{W} \qquad (3)$$

Where $W_L$ is the weight of the liver (g, ± 0.1 g) and W the total wet weight.

The mean oocyte diameter was assessed by measuring the longest (i.e. oocyte length, ± 0.1 mm) and shortest diameters (i.e. oocyte width, ± 0.1 mm) of 30 oocytes per female (n = 600 in total) using ImageJ (version 1.8.0_172 [64]). Oocyte ellipticity (OE) was calculated using the following formula:

$$OE = 100 * \frac{(L_o - W_o)}{(L_o + W_o)} \qquad (4)$$

Where $L_o$ is the oocyte longest length (mm) and $W_o$ the longest width perpendicular to $L_o$ (mm).

Otolith images were captured using a Sony® XCD-U100CR camera, with a dark background and under reflected light. Calibrated images were processed using ICY software (version 2.4.0.0 [65]). The age was estimated by counting the number of annual growth increments from the *nucleus* to the edge of the sagittal otolith. Length between the otolith *nucleus* and each growth increment (i.e. increment length) and between the *nucleus* and the edge (i.e. otolith radius) were also measured, always along the longest growth axis.

Since age is an integrative predictor, female life histories were also investigated. To do so, lengths-at-ages were retro-calculated from otolith measurements using the Biological Intercept model [66, 67]:

$$TL_i = L_{cap} + \left(R_i - R_{cap}\right) \frac{\left(L_{cap} - L_0\right)}{\left(R_{cap} - R_0\right)} \qquad (5)$$

Where $TL_i$ is the total length at age i, $L_{cap}$ the total length of the fish at capture, $R_i$ the distance between the *nucleus* and the last increment at age i, $R_{cap}$ the otolith radius at capture, $R_0$ is the radius of first otolith increment (i.e. at hatching; mean value: 14.75 μm) which corresponds here to the mean of values found in literature [39, 68–75] and $L_0$ the average length at hatching of larvae measured in this study. These back calculated data were then analyzed by fitting a von Bertalanffy growth model using a non-linear mixed effect model (following [67]). Von Bertalanffy growth parameters were calculated for each fish using Eq 6:

$$TL_i = TL_\infty \left(1 - e^{-K*(t-t0)}\right) \qquad (6)$$

Where: $TL_\infty$ is the asymptotic average total length, K is the Brody growth rate coefficient and $t_0$ is the theoretical age at which the average length is zero. Body growth patterns obtained in this study for the different females were compared to other *Clupeidae* (data extracted from FishBase [76]) with an auximetric plot ($\log_{10}$ plot of K versus $TL_\infty$). This procedure allowed us to verify the validity of our estimates.

Muscle RNA:DNA was evaluated for each female as it is an indicator of recent fish growth and condition [77, 78]. About 10 mg of fish muscle was sampled and placed in 200 μL of Tris-SDS (0.1 M NaCl, 50 nM Tris-HCl pH 7.8–8.5, 10 mM EDTA). Muscle was crushed with a

glass rod and total RNA and DNA were extracted following a protocol modified from [79]. Three glass beads were added per sample to disrupt tissue, and tubes were vortexed for 15 min, then centrifuged for 10 min at 4°C and at 8000 g. 360 μL of Phenol/Chloroform/Isoamyl alcohol 25:24:1 (pH 7.8–8.2 at 20°C) was then added and tubes were centrifuged at 4°C 10 min at 3800 g after vortexing for 5 min. The nucleic acid phase (top layer) was then transferred to a new Eppendorf tube. The tube containing phenol/chloroform/isoamyl alcohol was vortexed again for 5 min then centrifuged (4°C, 3800 g, 10 min), and the remaining upper layer was transferred to the nucleic acid tube to optimize the collection of DNA and RNA. Then, 180 μL of chloroform/isoamyl alcohol were added to the nucleic acid tube, which was then vortexed for 5 min and centrifuged (4°C, 3800 g, 5 min). The upper phase was transferred to a new tube to measure DNA and RNA contents. DNA and RNA concentrations were measured with a Qubit® fluorometer using the Biotium AccuGreen™ High Sensitivity dsDNA quantification kit and the Biotium AccuBlue® Broad Range RNA quantification kit, respectively, following manufacturer guidelines.

**2.4.2. Embryonic traits.** Fertilization rate, measured as the ratio between developed eggs (i.e. eggs that got fertilized and survived) and the total number of eggs deposited per plate, was evaluated for all plastic plates (four plates per family and per temperature regime; 20 families evaluated; total of plates: n = 240) under a binocular microscope (Olympus® SZX16, at 2–10× magnification) at one dpf (day(s) post-fertilization). Grey eggs with no sign of cell division or eggs which had stopped development following [80] were classified as unfertilized. Because of very low fertilization rates, numerous families were excluded after this check and only six families (one from the first batch, five from the second batch) were considered for further trait evaluation. Two plates per family were checked daily to follow daily development. Further traits were measured at two development stages [80]: eyed stage (i.e. when the eye is fully pigmented) and hatching. At eyed-stage, the mean between the longest and shortest diameters across the egg shell allowed evaluation of the mean egg diameter of 2–17 eggs per family and per temperature (variable n due to low survival rates from two families, particularly at 14°C; total n = 189). Survival rate was also estimated for all plates by evaluating the number of eggs that reached the eyed stage relative to the number of eggs that were fertilized (six families; four plates per family and per temperature regime; total of plates: n = 71; one plate excluded because of an initial null fertilization rate). Two more families (from the second batch) were excluded from analyses after the eyed-stage because of very low survival rates. Development rates until the eyed-stage and until hatching were calculated in terms of half-days post-fertilization for each plate and each family (variable n due to the exclusion of plates with null fertilization or null survival rate; n = 57 plates at eyed stage considering all temperatures, n = 34 plates at hatching considering all temperatures). Hatching rate was then estimated by evaluating the number of hatched larvae relative to the number of eggs that had reached eyed stage. Plates that were regularly manipulated to follow development (two plates per family) were excluded from the hatching rate assessment to avoid potential manipulation bias. Hatching rate was therefore evaluated for two plates per family and per temperature (two plates x four families x three temperatures; total n = 23 plates considering all temperatures, because one plate was excluded due to an initially null fertilization rate). Hatching was checked twice per day (morning, afternoon) and hatched larvae were sampled (globally 5–20 larvae/female/temperature), killed with an overdose of MS-222, transferred into 1.5 mL Eppendorf tubes with seawater and frozen at -80°C (methodology similar to [81]). Larvae were later thawed and photographed with an Olympus® SZX16 camera in order to measure (± 0.1 mm) total length (i.e. from the tip of the upper jaw to the end of the notochord), yolk sac height (mm), yolk sac length (mm) and myotomal depth (mm) using ImageJ software. In order to avoid a potential confounding effect from hatching date on key larval traits, only larvae from peak hatching (>50%) were

used in the measurement analyses. Yolk Sac Volume (YSV, mm$^3$) was calculated using the following formula:

$$YSV = \frac{\pi}{6} * YSL * YSH^2 \tag{7}$$

Where YSL is the yolk sac length (mm) and YSH is the yolk sac height (mm).

The newly hatched larvae were classified following [82]: stage 1a when the yolk sac height was lower than 2.5x the mytotomal depth and stage 1b when the yolk sac height was between 1x to 2.5x the myotomal depth. Finally, to assess larval dry weight, larvae were rinsed with distilled water and dried in Eppendorf tubes at -30˚C for one hour with a Christ® laboratory freeze-dryer Alpha-1-4-LCSplus. Dried larvae were then weighed to the nearest μg using a Sartorius® ME microbalance.

## 2.5. Statistical analyses

All analyses were performed in R 4.1.1 [83]. Statistical significance was assigned at α = 0.05. Because of the limited number of families and variable number of families/plates/eggs/larvae during development, analyses of the two objectives (i.e. temperature effect and potential relationships between female and embryo traits) were performed separately.

**2.5.1. Temperature effect.** Proportional traits (i.e. fertilization rate, eyed-stage survival rate, and hatching rate) were calculated per plate and plates were therefore considered as statistical replicates (n = 2–4 plates per family and per temperature regime). For other traits (i.e. mean egg diameter at eyed stage, larval length at hatching, myotomal depth at hatching, dry weight at hatching and yolk sac volume at hatching), measurements were made directly on individual eggs and larvae. These latter were thus considered as statistical replicates. Linear mixed models with normal distribution (R-package *lme4* [84]) were used to assess the impact of temperature (fixed effect) on continuous response traits (i.e. mean egg diameter at eyed stage, larval length at hatching, myotomal depth at hatching, dry weight at hatching and yolk sac volume at hatching). For traits expressed as proportions varying between zero and one (fertilization rate, eyed-stage survival rate, and hatching rate) and traits expressed as counted days (i.e. half-days to reach eyed stage and days to reach hatching), generalized mixed linear models were used following a binomial or a Poisson distribution, respectively. In initial models, family, batch and incubator were added as random factors. Plate was also added as a random factor for egg and larval traits analyses. Comparisons of AIC (Akaike Information Criterion) between all model variants (with/without fixed and random effects) and evaluation of variances of random factors allowed selection of the best model.

For models with normal distribution, homoscedasticity and normality of the residuals were checked graphically using predicted versus residual and Q-Q plots, and tested using the Breusch-Pagan and Shapiro-Wilk tests, respectively. If necessary, the response variable was log-transformed to respect assumption requirements (for myotomal depth and length at hatching). When there was no influence of random factors, one-way analysis of variance was used (ANOVA F test), followed by, if statistically significant, Tukey's HSD post-hoc tests to evaluate differences between temperatures. If at least one random factor was kept in the model, ANOVA was performed on the linear mixed model followed by, if necessary, a post-hoc test (estimated marginal means; R-package *emmeans* [85]) to compare temperature regimes. For all post-hoc tests, p-values were corrected using the Benjamini-Hochberg procedure.

For binomial or Poisson generalized linear mixed models, overdispersion was checked using the R-package *RVAideMemoire* [86]. Since there was evidence of overdispersion for all binomial models, quasibinomial distribution was used. Wald tests were performed to assess

the significance of the temperature factor, followed if necessary by estimated marginal means post-hoc tests. For all post-hoc tests, p-values were corrected using the Benjamini-Hochberg procedure.

**2.5.2. Relationships between female characteristics and embryo traits.** These preliminary analyses could not integrate paternal traits since pools of males were used for each fertilization. The attention was therefore focused on maternal traits, which are often considered as predominant (see introduction). Pearson's correlation analyses were used to assess relationships between female attributes. A multiple linear model was built for each embryonic trait to evaluate relationships with female traits (predictors, here TL, SL, W, CF, HSI, GSI, RNA:DNA, age, K, TL∞, oocyte mean diameter and OE). Models were only built for response traits in which a family or a batch random effect was significant in the previous section (i.e. fertilization rate, survival rate at eyed stage, mean egg diameter and hatching rate). Binomial generalized linear models were used for fertilization rate, survival rate and hatching rate while a normal distribution was used for mean egg diameter. Predictor variables were scaled for each model. Temperature was included as a predictor to take into account temperature effect on the response variable. Full models were built (i.e. with all predictors) and then variables were removed according to three criteria: (i) collinearity (using the R-package *car* [87]) by calculating GVIF (Generalized Variance Inflation Factor) and GVIF$^{1/2*Df}$ (with df: the number of degrees of freedom [88]) with a threshold set to 5, ii) identification of the parsimonious model (removal of non-significant predictors), and (iii) biological relevance according to the response trait considered and based on literature. For mean egg diameter, assumptions of homoscedasticity and normality of residuals were checked graphically, along with linearity and the absence of outliers (R-package *performance* [89]). For proportion variables, overdispersion was checked, which led to the use of quasibinomial distribution. Model outputs were checked using the R-package *jtools* [90].

# 3. Results

All male trait values are available in S3 Fig and a variability can be seen across male pools used to fertilize the different female eggs.

## 3.1. Temperature impact on embryo traits

Temperature had a statistically significant effect on eight functional traits (Table 2). Fertilization rate (Fig 1a) and mean egg diameter (Fig 1b) decreased with increasing temperature.

**Table 2. Impact of temperature on fertilization rate and embryonic traits of Downs herring (*Clupea harengus*).**
P-values in bold are statistically significant (statistical significance was assigned at α = 0.05). F: ANOVA F-test; $\chi^2$: Wald test; * indicates traits that were log transformed.

| Parsimonious model | F/$\chi^2$ | Df | P-value |
|---|---|---|---|
| Fertilization rate ~ temperature + (1\|family) | $\chi^2$ = 50.80 | 2 | <**0.001** |
| Mean egg diameter ~ temperature + (1\|family) + (1\|batch) | F = 28.82 | 2 | <**0.001** |
| Survival rate ~ temperature + (1\|family) | $\chi^2$ = 19.93 | 2 | <**0.001** |
| Number of half-days to reach eyed stage ~ temperature | $\chi^2$ = 26.25 | 2 | <**0.001** |
| Hatching rate ~ temperature + (1\|batch) | $\chi^2$ = 13.94 | 2 | <**0.001** |
| Length* ~ temperature | F = 1.42 | 2 | 0.24 |
| Myotomal depth* ~ temperature | F = 2.08 | 2 | 0.13 |
| Dry weight at hatching ~ temperature | F = 2.56 | 2 | 0.08 |
| Yolk sac volume ~ temperature | F = 3.04 | 2 | **0.05** |
| Number of half-days to reach hatching ~ temperature | $\chi^2$ = 32.97 | 2 | <**0.001** |
| Development stage at hatching ~ temperature | $\chi^2$ = 11.31 | 2 | <**0.05** |

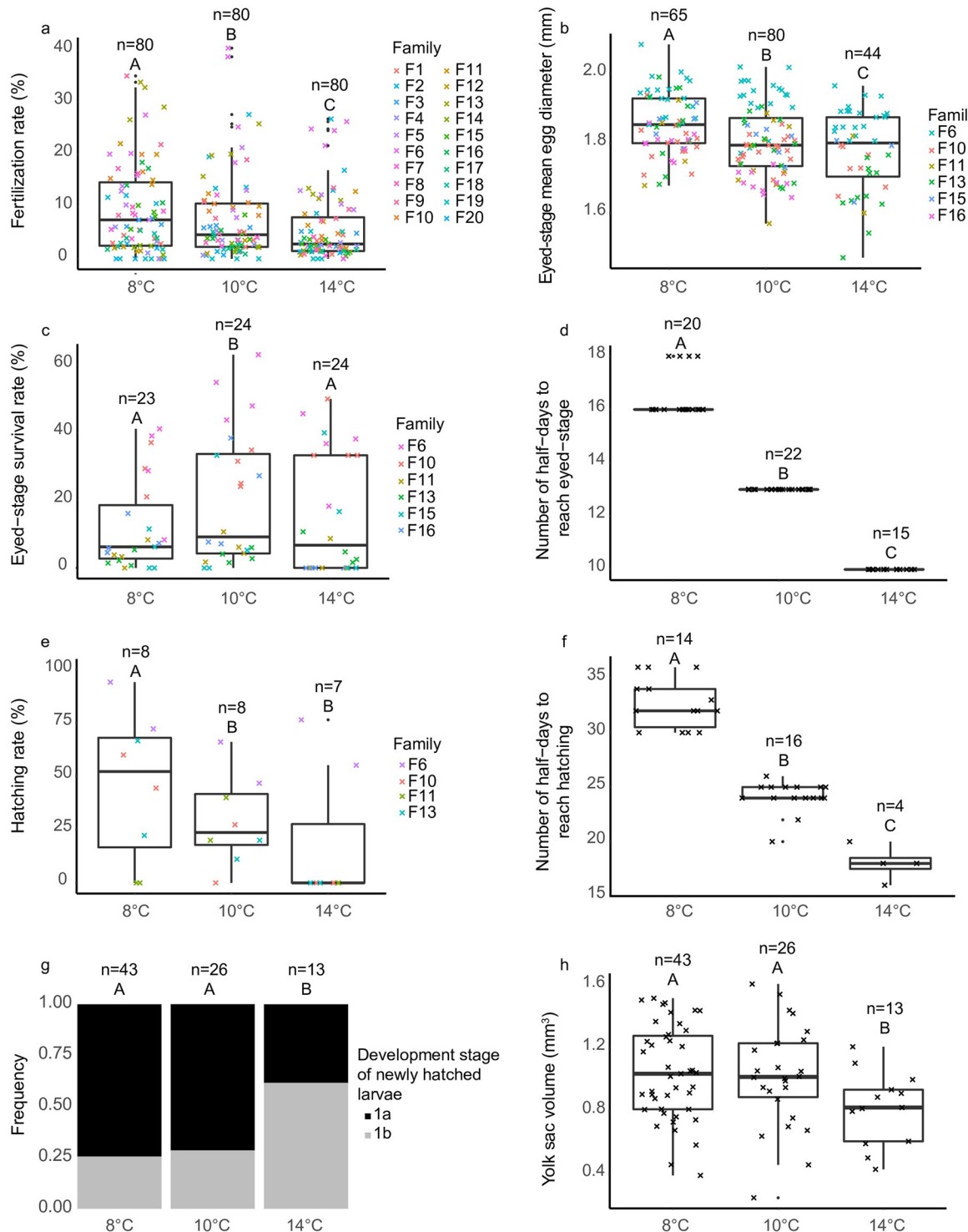

**Fig 1. Impact of temperature (8˚C, 10˚C and 14˚C) on Downs herring (*Clupea harengus*) a) fertilization rate, b) eyed-stage mean egg diameter, c) eyed-stage survival rate, d) number of half-days required to reach eyed-stage, e) hatching rate, f) number of half-days required to reach hatching, g) development stage of newly hatched larvae and h) yolk sac volume.** Letters in capital correspond to post-hoc results with different letters indicating statistical difference (statistical significance assigned at α = 0.05) between temperature treatments. Black cross points in graphs d, f and h correspond to sample values and black dots to boxplot outliers. When there was a significant parental effect in the model, points were colored relatively to families (graphs a, b, c and e).

Embryo survival rate at eyed-stage was significantly higher at 10˚C compared to 8˚C or 14˚C (Fig 1c). Development rates to reach eyed stage and hatching rate were different between temperatures (Fig 1d and 1f). Overall, larvae developed faster in terms of half-days post-fertilization when temperature was higher. In addition, hatching rate was lower at 10˚C and 14˚C compared to 8˚C (Fig 1e). Finally, there was a differentiation in the development stage frequency of newly hatched larvae among temperature regimes. More larvae reached stage 1b at 14˚C compared to the two control temperatures (Fig 1g). The yolk sac volume of newly hatched larvae was also larger for larvae at control temperatures (Table 2, Fig 1h). In contrast, no statistical difference was found for larval length at hatching (F = 1.42, df = 2, P = 0.24), larval myotomal depth (F = 2.08, df = 2, P = 0.13), or larval dry weight at hatching (F = 2.56, df = 2, P = 0.08) between temperature regimes (Table 2).

Family and/or batch random factors were important in models of four key traits: fertilization rate (range of values across families: 0–40.37%; S4 Fig), mean egg diameter (range of values: 1.47–2.08 mm), survival rate (range of values: 0–63.11%) and hatching rate (range of values: 0–94.11%) (Table 2; Fig 1).

## 3.2. Relationships between response variables and female traits

Trait measurements of females are available in Table 3 and life-history parameters (i.e. $TL_\infty$ and K) are represented in Fig 2. Our life-history estimates are among values previously reported for Clupeidae (Fig 2).

Overall, all regressions between female attributes and embryonic traits were statistically significant and explained a substantial proportion of variance (between 31 and 70%; Table 4).

**Table 3. Traits measured for the different Downs herring (*Clupea harengus*) females.** TL: Total Length; SL: Standard Length; W: Wet weight; FC: Fulton Condition index; GSI: Gonadosomatic index; HSI: Hepatosomatic index; sd: standard deviation.

| Female | Batch | TL (cm) | SL (cm) | W (g) | FC | GSI (%) | HSI (%) | Age (years) | Mean oocyte diameter (mm) | Mean oocyte ellipticity | RNA:DNA ratio |
|---|---|---|---|---|---|---|---|---|---|---|---|
| F1 | 1st | 27.5 | 24.5 | 209.7 | 1.01 | 27.71 | 0.33 | 7 | 1.5 | 0.02 | 2.09 |
| F2 | 1st | 29.5 | 26.5 | 227.0 | 0.88 | 26.65 | 0.26 | 5 | 1.6 | 0.03 | 2.36 |
| F3 | 1st | 28.0 | 25.2 | 226.7 | 1.03 | 33.08 | 0.31 | 7 | 1.6 | 0.03 | 1.78 |
| F4 | 1st | 29.7 | 27.0 | 232.3 | 0.89 | 23.63 | 0.82 | 7 | 1.7 | 0.02 | 1.88 |
| F5 | 1st | 31.0 | 28.0 | 269.6 | 0.90 | 22.70 | 0.26 | 7 | 1.6 | 0.02 | 1.97 |
| F6 | 1st | 29.2 | 26.5 | 249.2 | 1.00 | 31.38 | 0.60 | 8 | 1.6 | 0.03 | 1.34 |
| F7 | 1st | 29.0 | 26.2 | 220.1 | 0.90 | 15.26 | 0.59 | 6 | 1.6 | 0.04 | 1.71 |
| F8 | 1st | 28.0 | 26.3 | 197.9 | 0.90 | 26.07 | 0.91 | 7 | 1.6 | 0.03 | 1.50 |
| F9 | 1st | 30.0 | 26.5 | 224.2 | 0.83 | 24.84 | 0.18 | 8 | 1.7 | 0.03 | 2.27 |
| F10 | 2nd | 29.0 | 26.2 | 200.5 | 0.82 | 23.24 | 0.35 | 4 | 1.4 | 0.04 | 1.22 |
| F11 | 2nd | 26.2 | 23.5 | 157.2 | 0.87 | 26.08 | 0.19 | 4 | 1.5 | 0.03 | 1.96 |
| F12 | 2nd | 26.5 | 23.6 | 144.6 | 0.78 | 28.49 | 0.21 | 3 | 1.5 | 0.03 | 1.74 |
| F13 | 2nd | 27.5 | 25.5 | 208.7 | 1.00 | 32.19 | 0.29 | 3 | 1.6 | 0.03 | 2.16 |
| F14 | 2nd | 29.5 | 26.6 | 249.8 | 0.97 | 29.54 | 0.56 | 5 | 1.6 | 0.03 | 1.76 |
| F15 | 2nd | 25.4 | 22.7 | 135.6 | 0.83 | 20.21 | 0.22 | 3 | 1.6 | 0.03 | 1.41 |
| F16 | 2nd | 26.6 | 24.1 | 178.5 | 0.95 | 30.36 | 0.22 | 4 | 1.5 | 0.04 | 1.43 |
| F17 | 2nd | 26.4 | 23.7 | 160.2 | 0.87 | 29.34 | 0.19 | 4 | 1.5 | 0.02 | 1.80 |
| F18 | 2nd | 30.0 | 27.0 | 204.6 | 0.76 | 20.77 | 0.73 | 8 | 1.5 | 0.03 | 2.10 |
| F19 | 2nd | 27.4 | 24.3 | 145.9 | 0.71 | 21.18 | 0.21 | 5 | 1.5 | 0.04 | 1.52 |
| F20 | 2nd | 29.5 | 26.5 | 201.2 | 0.78 | 18.99 | 0.35 | 10 | 1.4 | 0.04 | 2.45 |
| Inter-female mean | | 28.3 | 25.5 | 202.4 | 0.89 | 25.60 | 0.39 | 5.75 | 1.6 | 0.03 | 1.82 |
| Inter-female Sd | | 1.5 | 1.4 | 36.9 | 0.09 | 4.70 | 0.22 | 1.98 | 0.1 | 0.005 | 0.34 |

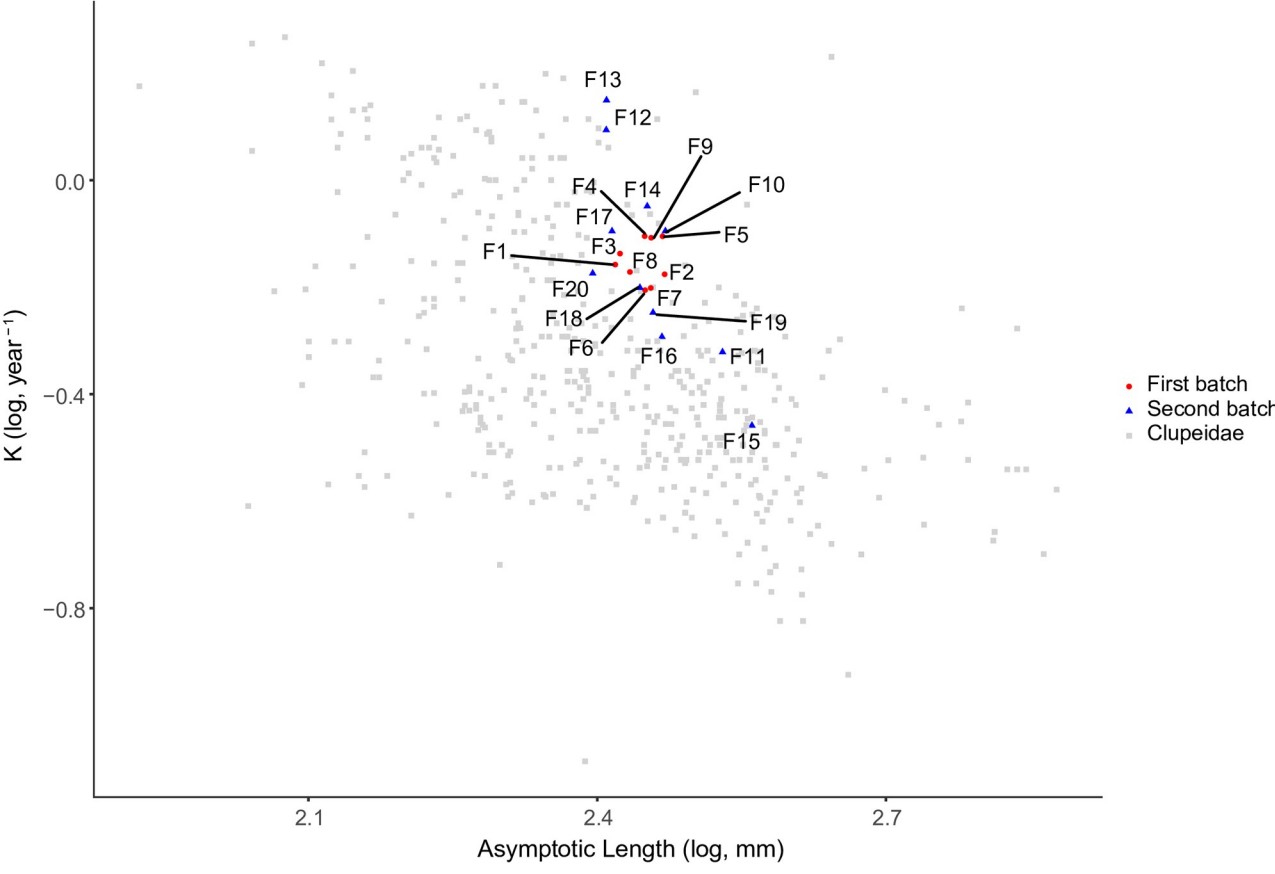

**Fig 2. Auximetric plot (K versus TL$_\infty$) for Downs herring (*Clupea harengus*) females (first batch in red, second batch in blue) and *Clupeidae* FishBase data (in grey).** Both axes are log$_{10}$ transformed.

Fertilization rate was higher for larger (SL, β = 10.10, *P* < 0.001) and in better condition (FC, β = 5.92, *P* < 0.001) females. It was also was positively affected by TL∞ (β = 2.48, *P* < 0.001); in contrast, it was lower for females with a higher HSI (β = -0.32, P < 0.05) and at high temperatures (β = -0.53 at 14°C, *P* < 0.001) (Table 4). Eyed-stage survival rate was higher in larger females (β = 25.45, *P* < 0.001) and at 10°C (β = 0.64, *P* < 0.001) while it was negatively affected by K (β = -2.49, *P* < 0.001) (Table 4). Finally, mean egg diameter and hatching rate were both higher in older females (mean egg diameter: β = 0.18, *P* < 0.001; hatching rate: β = 3.04, *P* < 0.001) (Table 4). In contrast, temperature had a negative effect on these two traits (mean egg diameter: β = -0.05 at 10°C and β = -0.10 at 14°C, *P* < 0.05; hatching rate: β = -1.31 at 10°C and -1.31 at 14°C, *P* < 0.01) (Table 4).

## 4. Discussion

Global sea surface temperatures are projected to increase between 0.7 and 3°C by the end of the century [91]. A large number of studies have previously investigated the impact of temperature on herring stocks, at larval (e.g. [92, 93]) or embryonic stages (e.g. [41, 42, 44]), but very few studies have focused on Downs herring (but see [57, 61, 94]). In order to better assess future impacts on Downs herring ELS, the effect of temperature on embryonic key traits was quantified across three temperature regimes. A strong impact of temperature was found on several key traits, particularly for the most extreme scenario, at 14°C. A strong variability in

**Table 4. Multiple regression results evaluating the relationships between embryonic traits and female trait predictors for Downs herring (*Clupea harengus*).** SL: standard length; FC: Fulton condition index; HSI: Hepatosomatic index; TL∞: Asymptotic average length; K: Brody growth rate coefficient; F: ANOVA F-test; $\chi^2$: Wald test; df: degrees of freedom; t: t-test; P: P-value. For temperature, which is qualitative, 8˚C is considered here as the comparison standard. P-values in bold are statistically significant (at α = 0.05).

| Response variable | Model variables | F/ $\chi^2_{(df,\ df\ residuals)}$ | P model | Pseudo- $R^2$ / Adjusted $R^2$ | Predictor estimates β | Standard error | t | P |
|---|---|---|---|---|---|---|---|---|
| Fertilization rate | SL | $\chi^2 = 2067.32_{(6,\ 232)}$ | <0.001 | 0.31 | 10.10 | 1.40 | 7.19 | <0.001 |
| | FC | | | | 5.92 | 0.74 | 7.99 | <0.001 |
| | HSI | | | | -0.32 | 0.15 | -2.17 | <0.05 |
| | TL∞ | | | | 2.48 | 0.43 | 5.78 | <0.001 |
| | 10˚C | | | | -0.15 | 0.13 | -1.19 | 0.24 |
| | 14˚C | | | | -0.53 | 0.14 | -3.64 | <0.001 |
| Eyed-stage survival rate | SL | $\chi^2 = 771.87_{(5,65)}$ | <0.001 | 0.70 | 25.45 | 2.09 | 12.15 | <0.001 |
| | K | | | | -2.49 | 0.29 | -8.47 | <0.001 |
| | 10˚C | | | | 0.64 | 0.17 | 3.64 | <0.001 |
| | 14˚C | | | | 0.12 | 0.20 | 0.60 | 0.55 |
| Mean egg diameter at eyed stage | Age | $F = 12.67_{(3,45)}$ | <0.001 | 0.42 | 0.18 | 0.04 | 4.62 | <0.001 |
| | 10˚C | | | | -0.05 | 0.02 | -2.07 | <0.05 |
| | 14˚C | | | | -0.10 | 0.02 | -4.13 | <0.001 |
| Hatching rate | Age | $\chi^2 = 90.00_{(3,22)}$ | <0.001 | 0.52 | 3.04 | 0.52 | 5.88 | <0.001 |
| | 10˚C | | | | -1.31 | 0.37 | -3.49 | <0.001 |
| | 14˚C | | | | -1.31 | 0.43 | -3.03 | <0.01 |

trait values across families was also showed for four key traits, fertilization rate, eyed survival rate, mean egg diameter, and hatching rate. Some female attributes are potentially useful for, at least partly, predicting embryo development and survival. In the following sections, we discuss these results regarding (i) temperature impact on key traits and variability across families, (ii) potential relationships between maternal characteristics and embryo traits, (iii) fertilization and survival rates and their determinants, and (iv) future perspectives from this work.

## 4.1. Impact of temperature on Downs herring key traits & variability across families

There was therefore a need to further explore the potential response of Downs embryos to warming, especially knowing that sea surface temperatures have already increased over time in the English Channel [95, 96].

The impact of temperature on fertilization rate is not often evaluated in experimental studies using wild dead fish as initial biological material. Indeed, the fertilization step is frequently performed at a control temperature before gradually increasing temperature afterwards (e.g. [44]). This method is used to guarantee the availability of initial biological material or to synchronize embryonic development to study the larval response afterwards. In the present study, there was a strong inter-family variability within each temperature regime (e.g. at 10˚C, variation between 0 and 40.4%), but the response to temperature also differed among families (e.g. fertilization rate decreased with rising temperature in F13 but no difference between temperatures was seen in F5; S4 Fig). As the response to warming differs among progenies, studying temperature impact on fertilization rate requires accounting for intra-stock unit variability. Yet, an overall decrease could still be seen with increasing temperature (Fig 1). Several mechanisms could explain such a decrease, for instance, an impact of temperature could exist on sperm motility [97–100]. Since fertilization rate is evaluated after 24h, this effect could also reflect the negative impact of high temperature on cell cleavage [101]. However, these

mechanisms require further study in Downs herring. The present results are different from those of Leo et al. [41], which showed a lack of temperature impact in Norwegian herring. This difference could reflect geographic differentiation, but it could also originate from a lack of intra-population variability consideration in that study. In this latter, three females were used and all progenies showed a lack of temperature effect, but we cannot exclude that other female progenies could have presented different response patterns.

It was also showed that hatching rate decreases with higher temperature, in agreement with Leo et al. [41] (but see divergent results in Peck et al [44]), although with a greater effect in our study. Hatching rates at 8°C and 10°C were within the range of previous experimental values [46, 50, 56, 81, 94, 102–105]. This decrease of hatching rate at high temperature was previously demonstrated for other species, such as *Clupea pallasi* [106] or *Inimicus japonicus* [107]. Similarly to fertilization rate, we found differences in hatching rate among families at the different temperatures. At 14°C, there was a negative impact of temperature on hatching rate of second-batch spawner offspring (F10, F11, F13; Fig 1; younger females which are potentially recruit spawners) while the impact was much reduced for the first-batch spawner progeny (likely repeat spawner [i.e. more experienced female which spawned in the previous years], F6; Fig 1). This effect could be linked to egg quality (i.e. ability of the egg to be fertilized and successfully develop into a normal embryo [99]) but, because of an insufficient number of females/families, this hypothesis requires further exploration.

Survival rate at eyed stage, contrary to the response seen in other key traits, was higher at 10°C compared to 8 and 14°C (Fig 1). However, survival rates were highly variable within each temperature regime and the temperature difference appears largely due to the variability seen among families (i.e. the embryos of some families showing higher survival rates at 10°C, Fig 1). Overall, survival rate is more variable among families than among temperatures. von Nordheim et al. [108] observed strong mortalities at higher temperatures (i.e. 11–13°C compared to 4–6°C) but this study was performed on Baltic herring. Survival rates may be linked to female attributes and egg quality [56] and/or thermal preferences could vary between stock units; no information is available, however, regarding the Downs component, and the comparison of thermal preferences between stock units is often limited by a lack of common methodology. The lack of strong mortality rate at 14°C compared to control temperatures throughout the experiment could reflect that there was no limitation in oxygen supply (i.e. oxygen demand, due to metabolic costs, rises when temperature increases). This was likely due to the capacity of the experimental system to maintain oxygen saturation levels above 100% in all treatments (Table 1). In nature, oxygen solubility decreases with rising temperature, posing a potential risk to organisms unable to seek more suitable areas, such as embryos present in benthic substrates. Moreover, temperature rise can also be associated in the wild with other stress sources such as increased occurrence of fungal spores or algal blooms [108, 109]. Therefore, further work needs to evaluate the combined impact of temperature and co-occurring stressors to reflect more accurately potential future consequences in the natural environment.

Temperature also affected growth traits. Mean egg diameter at eyed stage decreased with increasing temperature (Fig 1), which is congruent with Høie et al. [39] and might be related to differential growth rates because of temperature-mediated metabolic reactions. It has previously been shown that larvae hatching from larger eggs have a competitive advantage and better survival chances [30]. An additional study is required to assess if this difference at eyed-stage triggers differential growth trajectories across larval stages. The lack of temperature effect on larval length, larval dry weight and myotomal depth at hatching does not align with the eyed-stage mean egg diameter results, however larvae at 14°C are largely at stage 1b, with lower yolk sac reserves, while larvae at 8 or 10°C are mostly at stage 1a. Therefore, the comparison of larval characteristics at hatching is biased as they are evaluated at different development

stages; inter-temperature differences in larval traits could thus be seen at later development stages. Our results are consistent with those of Blaxter and Hempel [57] for Downs herring, which showed no body weight difference at hatching between two temperature regimes (8 and 12°C). In other stocks, a trend towards hatched larval length reduction with warming was assessed (e.g. [39, 41, 110]) but Peck et al. [44] found no difference in larval length at hatch between 8 and 14°C. Overall, variable patterns can be seen across the literature and herring stock units in response to warming. Additional morphological traits could be considered with bigger sampling sizes and additional methodologies such as deformity rate (e.g. using radiographs or staining methods), which could have consequences on later growth and survival from inefficient larval swimming abilities leading to lower efficiency to capture preys or avoid potential predators, or embryonic heart rate which would reflect metabolic activity [108, 111].

Globally, temperature seems to affect development rate more than growth rate. It has an inverse relationship with the development duration from fertilization to a specific stage/event (i.e. eyed-stage, hatching). The observed faster development at higher temperature is congruent with previous studies on Atlantic herring (e.g. [57, 112]). An effect of temperature on relative organogenesis timing has previously been demonstrated in Atlantic herring [80, 113–115]. In many fish, hatching enzymes have been shown to have lower activity at colder temperatures, extending the time spent inside the egg [116]. Variations in development speed have previously been shown across stock units and years relatively to temperature [32, 110]. The difference in development duration between temperatures could also be seen in larval stage at hatching and in yolk sac volume. There was a trend towards lower reserves available at hatching at 14°C (i.e. majority of stage 1b larvae at the highest temperature regime), which could be explained by faster development requiring more energy mobilization and therefore a faster consumption of yolk reserves. This is also congruent with the smaller mean diameter of eggs at eyed-stage at higher temperatures since smaller eggs provide less reserves to larvae at hatching [30, 57]. The present conclusions regarding the link between yolk sac reserves and temperature are contrary to those of Blaxter and Hempel [117] and Høie et al. [39] who found that larvae had a bigger yolk sac at higher temperatures (i.e. majority of stage 1a larvae at high temperatures). Those studies suggested a more efficient use of yolk reserves at lower temperatures since hatching occurred at a later stage of ontogeny. However, these studies were performed on Baltic and Norwegian herring stocks and could illustrate geographic differentiation in developmental response to temperature. Impact on yolk sac volume needs to be investigated further, alongside yolk sac consumption efficiency. Indeed, higher temperatures during herring egg incubation could have significant impacts if prey availability is limited (i.e. higher risk of starvation because of lower reserves available at hatching). In addition, higher prey availability could be required after yolk absorption at warmer temperatures [118]. In contrast, faster development could allow earlier feeding which can be an advantage over late hatching larvae and against predation if they reach bigger sizes more rapidly. Importantly, temperature experienced during embryogenesis can also influence fitness later in the life cycle (e.g. growth, age at maturation, reproductive investment [29]). The assessment of temperature effects should therefore be continued for the subsequent life stages. In particular, it would be interesting to assess the impact of warming on breeders since they also present a narrower thermal tolerance [25]. Temperature could influence reproductive dynamics [112] and negatively impact gametogenesis and egg quality (e.g. negative impact on fertility, chorion integrity, sperm DNA integrity or velocity; see review in [101]). Additional studies are thus required to better evaluate potential future implications in the wild for Downs herring.

Based on the present results, warming could have negative consequences for Downs herring recruitment because of lower fertilization rate, lower hatching rate and faster development at higher temperatures. The lower performances seen at the highest temperature could

be a first insight of adaptability limitation. The strong inter-family variability of key trait response to temperature illustrates, however, a potential plastic response to warming. It suggests that within one stock unit there is a range of potential responses to warming, which could be indicative of an adaptive potential to warming of Downs herring, but this hypothesis requires further evaluation with a larger number of families. It appears crucial to account for intra-population variability when investigating the response of a stock component to stressor(s). Downs herring being located during spawning at the southern boundary of the Northeast Atlantic range, they could adopt other strategies to avoid warmer temperatures such as northward migration or changes in spawning timing or grounds [119, 120]. Shifts in spawning season would be particularly interesting to further investigate. They would be congruent with the hypothesis of a match between spawning temperature and embryo thermal preference [13]. Phenological shifts in spawning due to changes in temperature have been shown for a wide range of taxa (see examples in [101, 121]), including Baltic herring [122]. Overall, shifts in spawning timing and faster development across embryogenesis could potentially trigger an even stronger mismatch with planktonic prey sources. This could have profound consequences on the North Sea ecosystem dynamics, as herring is at the center of the trophic chain (bottom-up regulation of seabirds and top-down regulation of zooplanktonic species [123]).

## 4.2. Parental effects and first insights on relationships between maternal attributes and embryo traits

Inter-family variability was found for four key traits (i.e. fertilization rate, eyed-stage survival rate, eyed-stage mean egg diameter and hatching rate; Fig 1; Table 1). The lack of parental effect for some traits, such as larval morphological traits, could be due to the limited number of larvae and families studied (but see a lack of female effects demonstrated in other stocks, e.g. [50]). This study represents thus a preliminary work towards the evaluation of parental effects and further studies should be performed to validate/refute our first observations.

Due to our experimental design, we evaluated an overall parental effect, where paternal traits could not be fully isolated and studied. However, it is often considered that maternal effects are particularly important and often predominant [39, 52–55, 124]. This work initiated therefore the identification of potential relationships between maternal attributes and embryo traits. More precisely, due to the design of our experiment, maternal effects could not be separated from genetic female effects. We therefore investigated the potential global female effect (i.e. contribution of both non-genetic and genetic effects).

Inter-family variability could reflect differences in egg quality. A good egg quality refers to an embryo that survives and develops into a viable larva [99]. Decades of research in the aquaculture field have allowed identification of various quality indicators such as egg morphology (e.g. egg size, symmetry) or egg content (e.g. lipid and protein contents) [125, 126]. Egg size usually reflects female investment in offspring and is often considered a fitness determinant in many fish species [55]. Other traits linked to offspring viability, such as fertilization success, survival rate at eyed-stage or hatching rate, are usually used to assess quality [99, 126, 127]. With the present data, it can be seen, for instance, that the progeny F6, belonging to the first batch, had one of the highest fertilization rates, the highest mean egg diameter, a higher survival rate at eyed-stage and a better hatching rate across temperatures (Fig 1, S4 Fig) compared to embryos from other families. Overall, eggs from this specific family present a better quality than those from other families originating from the second batch. This specific family also appears as less affected by temperature increase (Fig 1, S4 Fig) and there is thus further work to lead on the link between offspring quality and temperature response. Overall, there is

therefore an interest in the identification of predictors to identify good quality females prior to embryonic assessment.

In fish species, egg quality is affected by several attributes such as fish age, size or body condition, which is partly linked to underlying parental nutrition status and environmental conditions experienced during sexual maturation [99, 127, 128]. In this study, we investigated whether embryonic survival and developmental success could be linked to specific female attributes across temperature regimes. Some of these traits, such as FC or GSI, represent proxies for female condition/reproductive investment. Some female attributes do not appear as useful egg quality predictors. The ratio RNA:DNA, for instance, does not appear as relevant but it is an indicator that has not often been used in breeders (e.g. [77]) and that is usually employed for larvae quality assessment (e.g. [129]). On the contrary, some female traits are strongly correlated and are therefore interchangeable (e.g. W and FC / W and SL / W and age / GSI and FC / oocyte diameter and W positively correlated, K and $TL_\infty$ / oocyte ellipticity and mean oocyte diameter negatively correlated; S5 Fig). Overall, it could be stated that higher fertilization rate, eyed survival rate, mean egg diameter and hatching rate were obtained in older, larger and/or in better condition females (Table 3). This is congruent with the big old fat fecund female fish (BOFFFFs) hypothesis [130]. Age-related egg quality variations are well-known from literature, especially when comparing recruit and repeat spawners (see e.g. [57]; other species examples in [101]). In most cases, older females produce eggs of higher quality than younger females [53, 131]. Several criteria could explain age-related variations in fish species, such as younger females having lower condition, allocating less energy to spawning because of growth demands, experiencing poorer habitats or exhibiting a partially activated endocrine system [101, 132]. Egg composition in terms of quality and quantity (e.g. cholesterol, fatty acids, free aminoacids profiles) could be an interesting predictor to study since it reflects the availability and quality of food and the subsequent energy budget of the female. In small pelagic fishes, the amount of lipid reserves constitutes a good indicator since lipids are stored into the ovaries during oogenesis and are the main constituent of the egg yolk [133]. This criterion would be useful to explore underlying mechanisms defining maternal effects (see e.g. [56, 134]). Looking at other significant predictors, the positive relationship between fertilization rate and $TL_\infty$ and the inverse relationship between K and eyed survival rate show that life history is also an important factor to consider. These results might illustrate different strategies with some females investing more energy into reproduction than growth. Finally, a negative relationship was highlighted between HSI, a proxy for energetic reserves, and fertilization rate. This is consistent with the fact that liver provides lipids for gonadal development [135, 136]. In that way, hepatic energetic reserves of females are low during the last step of the reproductive cycle [137].

Globally, the present results provide first insights on potential egg quality indicators and could help to optimize fertilization and embryonic development for future experiments. Indeed, despite the challenges, working with dead biological material is practical, and improved female selection could greatly improve chances to obtain a sufficient number of viable eggs. A complementary experiment with a larger number of females, which would integrate additional predictors, such as egg composition and fertility, is necessary to validate/refute these preliminary conclusions. In addition, other criteria could also influence reproductive performance such as exposure to persistent organic pollutants [138], even though this could be partly reflected by condition index. A more exhaustive assessment of female state should thus be led. Repeating this experiment is also essential since there could be a potential noteworthy impact of conditions experienced by parents/grandparents (i.e. transgenerational effects [139]), triggering a potential inter-annual variability in egg quality. Indeed, the energy budget of females depends on nutrition but also on environmental abiotic

factors. These latter affect the neuroendocrine system that controls energy expenditure and metabolism of the breeders [133]. In this way, thermal regimes experienced by parents affect early development and contribute to shape future juveniles fitness, which could partly explain differences between batches. This could trigger inter-annual variations in fertilization and survival rates, as previously shown in Baltic Sea herring [56]. Overall, this study represents a first step in the study of parental effects, and likely mostly maternal effects, in Downs herring. These results may have implications for fisheries management; indeed, fisheries act as a selective pressure that often leads to remove larger/older individuals, which can eventually lead to truncated age structure of the stock [130, 140–142]. Female attributes can greatly influence the quantity and quality of offspring, and therefore the stock recruitment success. In this study, since older/bigger females produce eggs of better quality, it can be argued that the preservation of Atlantic herring stock structure (large age range), and underlying genetic diversity, is crucial for recruitment.

Finally, while this study was focused on female traits, male effects cannot be excluded. Milt from three males was used to fertilize eggs of each female in order to minimize potential male effects and optimize fertilization success. Looking, however, at morphological characteristics of males, there is a wide variability in male condition, particularly when comparing age, wet weight or gonadosomatic index (S3 Fig). In addition, eggs could be more easily fertilized by the sperm of some specific males [143] and potential paternal effects could therefore play a role in embryo trait expression. Variability seen in embryonic traits originating from different females might also, therefore, be partly due to male effects, through a global family effect. The importance of paternal effects, which are genetic effects (i.e. contribution of the milt is DNA), was previously demonstrated for several species (e.g. [53, 144, 145]), including Atlantic herring [50, 146]. Our models assessing relationships between female attributes and embryo traits only explain a part of the variability. Integrating male quality traits (e.g. sperm mobility, sperm velocity, DNA integrity [101]) could allow to increase the accuracy of the predicting models. However, other studies also reported a lack of paternal effect for some traits [39, 46]. Moreover, no correlation was found between the male characteristics (e.g. size, GSI, sperm characteristics) and fertilization rate in another reproductive component from the Irish Sea [147]. The importance of male effects remains therefore uncertain for Atlantic herring. Ruling out paternal effects in future studies would require fertilizing all eggs with one single male, which was not technically possible in this study because of the fertilization protocol used (i.e. too long time between the first plate fertilization and the last one as the sperm gets activated as soon as it gets in contact with the first wet eggs). Future studies integrating specific male quality indicators (e.g. sperm motility, DNA integrity) are necessary to improve our understanding of the contribution of males to their offspring quality.

### 4.3. Fertilization and survival rates: Multifactorial traits

A striking feature of our study is the low fertilization and survival rates, which resulted in a large decrease of the number of families studied during embryonic development. Low fertilization rates (<40%) in standardized experimental conditions have previously been reported for Atlantic herring (e.g. [46, 103, 105, 110, 147]). Still, higher fertilization rates have been estimated, particularly in the case of Baltic Sea herring (e.g. [108]), as well as higher survival rates for other stock components [44, 56, 104, 108]. The low fertilization and survival rates obtained in this study could result from a combination of ecological and methodological features, as discussed below.

Firstly, a low rate of fertilization could be specific to Downs herring, as very limited information is available for this stock component. The direct comparison of Down herring results

with other stock components (e.g., Baltic Sea herring) is limited because of the divergent life-history strategies and experienced environmental conditions. In addition, North Sea herring has experienced below-average recruitment since 2015 despite being exploited at maximum sustainable yield level. This pattern would be congruent with low fertilization and survival rates of the Downs component, which has markedly increased in recent years, contributing to 25% of the total North Sea herring spawning stock biomass in 2020 [148]. To our knowledge, no information is available regarding fertilization success in the field or subsequent survival of embryos. Two experimental studies, however, previously reported higher average fertilization rates: 42% in Joly et al. [63] and 75% in Illing et al. [149]. In these studies, breeders were sampled earlier in the spawning season in the first case (i.e. mid-November) or freshly captured fish were used in the second case. Higher average fertilization rate from Joly et al. [63] could be linked to fluctuating egg quality across the spawning season. In Joly et al. [63], females were sampled in November 2018 and could correspond to older females providing better quality eggs. No information regarding age is available in Joly et al. [63], but females used therein were larger than fish collected in our first batch (beginning of December). The variation in fertilization rate could also reflect inter-annual variability in egg quality and/or spawning timing. Indeed, looking back at water temperature in the eastern English Channel, Downs herring experienced slightly colder temperatures (1°C less) in 2018 compared to 2021 at the beginning of the spawning season [62]. This difference could have delayed maturation and spawning timing in 2021. Mature individuals were identified using classical morphological criteria (e.g. eggs hydrated, fluent males) but it is possible that those criteria are not sufficient to select ready-to-spawn individuals among mature fish. This reinforces the need to identify better proxies for fertilization success, which we initiated in this study. Second, the freshness of the biological material could also be limiting since the breeders in our study, in contrast with Illing et al. [149], were obtained dead from commercial fishing vessels and no information was available on the exact capture time, except that biological material originated from the last haul before returning to harbor. Although fish were collected after the catch had been landed on the deck and eggs were fertilized rapidly, time post-capture and the effect of rapid temperature change between ice-storage and seawater could both have substantial impacts on fertilization success and subsequent embryonic survival. Indeed, delayed fertilization after ovulation can lead to ova ageing and ultimately to over-ripening of the eggs [150]. Over-ripening phenomenon has been shown to be detrimental to egg quality and to impact fertilization success but also developmental success (e.g. lower survival rate, higher deformity rate; see [150] and references therein). A better solution would consist of stripping freshly captured fish and perform fertilization at sea using incubation boxes in which temperature can be controlled [125]. This method involves large logistic and financial resources, however, and even in this ideal scenario, fertilization rates could be biased since stripped and artificially fertilized eggs can be damaged in the process compared to naturally spawned eggs. Collecting dead fish from fishers represents a practical alternative and, as previously mentioned, the development of egg quality proxies will allow selecting the best females to optimize fertilization success and embryonic survival. Finally, as mentioned previously, male effects could partly determine fertilization success and it is possible that males used here for fertilization trials were not of best quality. In future studies, male effects should be considered and sperm quality should be evaluated (e.g. sperm motility). Overall, several ecological and methodological factors could have impacted fertilization success and subsequent survival rates. These factors do not, however, affect the differences seen across temperature regimes since eggs from each family were incubated in each temperature regime, and all abiotic factors, except for temperature, were the same across rearing systems (Table 1).

### 4.4. Perspectives

Overall, despite the low number of families, the present results suggest that some female attributes appear as important determinants for reproductive success, as previously shown in other Atlantic herring stocks (e.g. [39, 56, 57]), and also in other species [52, 134, 151, 152]. We also showed some differences in the response to temperature regimes across families (e.g. for fertilization rate) and this potential plasticity should be further explored by considering a higher number of families. The present results may not be representative of the reproductive component as a whole and the natural intra-population variability was likely under-estimated, but this work represents an important first step towards the assessment of responses to climate change mediated by potential parental effects. The present results will allow to explore parental effects in further detail by: (i) considering additional attributes (e.g. egg composition) to find additional reliable estimates for the production of eggs and larvae in recruitment assessments, (ii) thoroughly exploring the different characteristics and offspring quality of recruit vs repeat spawners (i.e. variability across spawning season), and (iii) assessing offspring quality variability across years. Future evaluations should also integrate potential male effects to disentangle properly maternal from paternal effects. Regarding the climate change impact, several further research directions also appear to be important to explore. We here assessed the impact of rising temperature using three regimes with constant temperatures. Even more than rising temperature, an important stress factor to study is the increase in extreme event frequency [91]. We therefore believe that an essential further step consists of investigating the impact of temperature variations. Finally, our study constitutes a single stressor, focused on temperature. This study should be extended to multi-stressor experiments by considering additional sources of stress such as acidification, hypoxia, salinity, UV changes, or concentration of heavy metals or contaminants [24, 111, 153].

## Supporting information

**S1 Fig. Temperature recorded from a Marel Carnot buoy (code 6200443) in front of Boulogne-sur-Mer between mid-November 2021 and mid-January 2022 (Coriolis data; Lefebvre, 2015).**
(PDF)

**S2 Fig. Experimental set-up.**
(PDF)

**S3 Fig. Morphological characteristics from pools of males (n = 3 per pool) used for the fertilization of eggs from the different females.**
(PDF)

**S4 Fig. Boxplots of fertilization rate across the different families.** Colors correspond to the three temperature scenarios: 8˚C (blue), 10˚C (green) and 14˚C (red).
(PDF)

**S5 Fig. Correlation plot of female attributes.** Positive correlations are indicated in red while negative correlations are indicated in blue. Color intensity and circle size are proportional to correlation coefficients.
(PDF)

## Acknowledgments

This work would not have been possible without the help of the FROM Nord (*Fonds Régional d'Organisation du Marché du poisson*) for providing fish for pre-trials and of the CME

(*Coopérative Maritime Etaploise*, Boulogne-sur-Mer) for furnishing biological material for experiments. Authors thank the Nausicaá aquarium (*Centre national de la Mer*) for hosting our experiment and more particularly Dominique Mallevoy and Florent Kapps for logistic support. Authors also acknowledge the laboratory of Oceanology and Geosciences of ULCO University and Sebastien Monchy for logistic and scientific supports for DNA and RNA extractions. Finally, authors are grateful to Léa Joly for her help with experimental design, Valérie Lefebvre and Guillaume Lescoute for their help with experiment preparation and egg fertilization and to Thibaut Kersaudy, Geoffrey Bled Defruit and Solene Telliez for extraction and ageing data from the otoliths.

## Author Contributions

**Conceptualization:** Lola Toomey, Carolina Giraldo, Christophe Loots, Kélig Mahé, Paul Marchal, Kirsteen MacKenzie.

**Formal analysis:** Lola Toomey.

**Funding acquisition:** Carolina Giraldo, Christophe Loots, Paul Marchal.

**Investigation:** Lola Toomey, Kirsteen MacKenzie.

**Methodology:** Lola Toomey, Carolina Giraldo, Christophe Loots, Kélig Mahé, Paul Marchal, Kirsteen MacKenzie.

**Resources:** Carolina Giraldo, Christophe Loots, Kélig Mahé, Paul Marchal, Kirsteen MacKenzie.

**Supervision:** Carolina Giraldo, Christophe Loots, Kélig Mahé, Paul Marchal, Kirsteen MacKenzie.

**Validation:** Lola Toomey.

**Writing – original draft:** Lola Toomey.

**Writing – review & editing:** Carolina Giraldo, Christophe Loots, Kélig Mahé, Paul Marchal, Kirsteen MacKenzie.

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
