## [Decision Letter · Decision Letter 0]

14 Feb 2023

PONE-D-22-34768Impact of temperature on Downs herring embryonic stages: first insights from an experimental approachPLOS ONE

Dear Dr. Lola,

Thank you for submitting your manuscript to PLOS ONE. After careful consideration, we feel that it has merit but does not fully meet PLOS ONE’s publication criteria as it currently stands. Therefore, we invite you to submit a revised version of the manuscript that addresses the points raised during the review process.

We look forward to receiving your revised manuscript.

Kind regards,

A. K. Shakur Ahammad, PhD

Academic Editor

PLOS ONE

Journal Requirements:

"This work was financially supported by the European Union (ERDF), the French State, the French Region Hauts-de-France and Ifremer, in the framework of the project CPER MARCO 2015-2021. "

"This work was financially supported by the European Union (ERDF), the French State, the French Region Hauts-de-France and Ifremer, in the framework of the project CPER MARCO 2015-2021. The funders had no role in study design, data collection and analysis, decision to publish, or preparation of the manuscript."

Additional Editor Comments:

Dear Authors,

The authors have evaluated the “Downs herring embryonic stages: first insights from an experimental approach, in relation to temperature”. This work has very good practical implications in increasing the seed production of herring fish. This work has been planned and executed well and the tables & figures are well represented.

Generally, this research is original and a nice contribution in the field of embryology. The Downs herring is an very important fish species for mariculture. Some of the important recent work may be added in the Manuscript.

The paper is well written and adds to existing literature on the subject. The introduction is well composed and has been developed on the right lines. All portion of your methodology is clear. The results have been presented well. The discussion has been well brought out but need to address the reviewer comments to justify your design and work. References are as required. There are some minor revision should be addressed. Please find the all reviewer comments through downloading the reviewed file.

With regards

Academic Editor

A. K. Shakur Ahammad

Reviewers' comments:

Reviewer's Responses to Questions

**Comments to the Author**

1. Is the manuscript technically sound, and do the data support the conclusions?

Reviewer #1: Yes

Reviewer #2: Yes

Reviewer #3: Yes

2. Has the statistical analysis been performed appropriately and rigorously? 

Reviewer #1: Yes

Reviewer #2: Yes

Reviewer #3: Yes

3. Have the authors made all data underlying the findings in their manuscript fully available?

Reviewer #1: Yes

Reviewer #2: Yes

Reviewer #3: Yes

4. Is the manuscript presented in an intelligible fashion and written in standard English?

Reviewer #1: Yes

Reviewer #2: Yes

Reviewer #3: Yes

5. Review Comments to the Author

Reviewer #1: Dear authors,

I have performed the review of your submitted work entitled "Impact of temperature on Downs herring embryonic stages: first insights from an experimental approach".

Overall, the submitted work is based upon a pertinent subject (Ocean warming effect on Herring embryonic development), allied to a detailed analyses of experimental data.

The work itself provides data that is relevant to move beyond the state of the art of the addressed topics. The significance of contents, scientific soundness, and overall interest to peers is high, altough minor revisions are in order, which could improve the overall output of what was done.

Please find attached, the submitted manuscript with comments included.

Sincerely yours,

The reviewer

Reviewer #2: The manuscript entitled “Impact of temperature on Downs herring embryonic stages: first insights from an experimental approach” is a well-written record of the impact of temperature on Downs herring embryonic stages in a practical way. The MS explains a scientifically planned research work, and the results were expressed with sufficient data. The research is a piece of novel information for the scientific community to understand the dynamics of the impact of temperature on Downs herring embryonic stages, which is much more relevant today. Even though the MS is written well, there are a few suggestions to improve MS. With below mentioned minor corrections; the MS may be considered for publication in the journal.

Title: The scientific name can be included in the title.

Abstract: Some of the values/statistics can be used in the abstract to explain the results of the present study to increase clarity. A sentence can be added at the end to demonstrate the relevance of the results or as a conclusion with the outcome of the study.

Introduction: I felt the introduction is very elaborate and needs to be concise. It can be started with the background information, research problem, present status of the research in that line, and what the present study aims to unravel.

Materials and Methods: Properly written with all required details.

Results and Discussion: Written well with good data representations.

Reviewer #3: This article aims to evaluate the effect of different water temperatures on the sensitive embryonic period and investigate potential parental effects on embryonic traits. The Ms in general presents well written and organized. For this reason, I think that after a minor revision this paper has to be published soon. Some comments are listed below:

Specific comments:

Line 175-176: All animal experiments should comply with the ARRIVE guidelines. It seems that all breeders were killed for GSI and muscle RNA:DNA studies, please revise it.

Line 223: Total Length, please use in lower case.

Line 282: Same as above. Sac Volume, please use in lower case.

Table 1: The authors should add explanations about experimental days under each temperature regime. To my knowledge, usually incubation time is given at half the day of yours (Blaxter, 1968).

2.5. Statistical analyses: Statistical measures such as SD or SE should be identified.

6. PLOS authors have the option to publish the peer review history of their article (what does this mean?). If published, this will include your full peer review and any attached files.

Reviewer #1: No

Reviewer #2: No

Reviewer #3: No

---

## [Author Response · Author response to Decision Letter 0]

20 Mar 2023

Editorial comments: 

Ed#1: “1. Please ensure that your manuscript meets PLOS ONE's style requirements, including those for file naming. The PLOS ONE style templates can be found at https://journals.plos.org/plosone/s/file?id=wjVg/PLOSOne_formatting_sample_main_body.pdf and https://journals.plos.org/plosone/s/file?id=ba62/PLOSOne_formatting_sample_title_authors_affiliations.pdf”

Author’s response: We checked the manuscript and modified it following requirements.

Ed#2: “2. We note that the grant information you provided in the ‘Funding Information’ and ‘Financial Disclosure’ sections do not match. When you resubmit, please ensure that you provide the correct grant numbers for the awards you received for your study in the ‘Funding Information’ section.”

Author’s response: Modifications were made.

Ed#3: “3. Thank you for stating the following in the Acknowledgments Section of your manuscript: "This work was financially supported by the European Union (ERDF), the French State, the French Region Hauts-de-France and Ifremer, in the framework of the project CPER MARCO 2015-2021." We note that you have provided funding information that is not currently declared in your Funding Statement. However, funding information should not appear in the Acknowledgments section or other areas of your manuscript. We will only publish funding information present in the Funding Statement section of the online submission form. Please remove any funding-related text from the manuscript and let us know how you would like to update your Funding Statement. Currently, your Funding Statement reads as follows: "This work was financially supported by the European Union (ERDF), the French State, the French Region Hauts-de-France and Ifremer, in the framework of the project CPER MARCO 2015-2021. The funders had no role in study design, data collection and analysis, decision to publish, or preparation of the manuscript." Please include your amended statements within your cover letter; we will change the online submission form on your behalf.

Author’s response: The funding statement is correct. We removed this statement from the Acknowledgements section.

Ed#4: “4. Please review your reference list to ensure that it is complete and correct. If you have cited papers that have been retracted, please include the rationale for doing so in the manuscript text, or remove these references and replace them with relevant current references. Any changes to the reference list should be mentioned in the rebuttal letter that accompanies your revised manuscript. If you need to cite a retracted article, indicate the article’s retracted status in the References list and also include a citation and full reference for the retraction notice.”

Author’s response: The reference list was checked.

Additional editor comments:

Ed#5: The authors have evaluated the “Downs herring embryonic stages: first insights from an experimental approach, in relation to temperature”. This work has very good practical implications in increasing the seed production of herring fish. This work has been planned and executed well and the tables & figures are well represented. Generally, this research is original and a nice contribution in the field of embryology. The Downs herring is an very important fish species for mariculture. Some of the important recent work may be added in the Manuscript. The paper is well written and adds to existing literature on the subject. The introduction is well composed and has been developed on the right lines. All portion of your methodology is clear. The results have been presented well. The discussion has been well brought out but need to address the reviewer comments to justify your design and work. References are as required. There are some minor revision should be addressed. Please find the all reviewer comments through downloading the reviewed file..”

Author’s response: We thank you for your recommendations. We responded to each of the reviewers’ comments and modified accordingly the manuscript.

Reviewer 1

R1#1: “I have performed the review of your submitted work entitled "Impact of temperature on Downs herring embryonic stages: first insights from an experimental approach". Overall, the submitted work is based upon a pertinent subject (Ocean warming effect on Herring embryonic development), allied to a detailed analyses of experimental data The work itself provides data that is relevant to move beyond the state of the art of the addressed topics. The significance of contents, scientific soundness, and overall interest to peers is high, altough minor revisions are in order, which could improve the overall output of what was done. Please find attached, the submitted manuscript with comments included.”

Author’s answer: We thank you for your detailed evaluation and we respond to all your specific comments below.

R1#2: “Suggestion: Maybe "settings" instead of "regimes"

Author’s answer: We removed regimes to simplify.

R1#3: “Suggestion: Maybe "Overall, ..." instead of "Global"

Author’s answer: Modification was made.

R1#4: “Suggestion: Maybe "observed" instead of "demonstrated"

Author’s answer: Modification was done.

R1#5: “Suggestion: Maybe "(eyed stage)" instead of "at eyed stage"”

Author’s answer: We kept the precision of the eyed stage because mean diameter could be measured at any time across development. Eyed stage is yet considered as a good comparison point across temperature scenarios since it is considered as a development stage.

R1#6: “Suggestion: Maybe "observable" instead of "highlighted"”

Author’s answer: We replaced “highlighted” by “observed”.

R1#7: “Suggestion: It would be more pertinent to include "ocean warming" instead of "climate change" or just simply "temperature"”

Author’s answer: You are right, we replaced it by ocean warming (temperature being already in the title).

R1#8: “Suggestion: maybe include "(IPCC)" after the highlighted text.”

Author’s answer: Modification was done.

R1#9: “Suggestion: maybe "ocean" instead of "water".”

Author’s answer: Modification was made.

R1#10: “Suggestion: maybe "a sea surface temperature (SST) increase" instead of "sea surface warming".”

Author’s answer: Modification was made but we did not add the acronym since it is not further used in the text.

R1#11: “Suggestion: maybe "climate change stressors (e.g. ocean warming)" instead of "climate change". Suggestion: maybe "could potentially" instead of "could".”

Author’s answer: Modifications were made.

R1#12: “Suggestion: maybe "ocean warming" instead of "warming".”

Author’s answer: The text was modified.

R1#13: “Please include a reference to sustain the underlying info of the sentence”

Author’s answer: A reference was added (Boyd & Brown, 10.3389/fmars.2015.00009).

R1#14: “Suggestion: I would use "drivers" instead of "stressors", once to some species, some could be stressors, while for others they could not pose or induce a stress response.”

Author’s answer: You are right, we modified the text.

R1#15: “Suggestion: I would use "while providing" instead of "and they provide".”

Author’s answer: Modification was done.

R1#16: “Suggestion: I would add afterwards "(ELS)" and use it subsequently throughout the text”

Author’s answer: Modification was done and we replaced it along the text.

R1#17: “Suggestion: I would use "fosused on the" instead of "investigated".”

Author’s answer: Modification was done

R1#18: “Suggestion: I would use "was" instead of "is".”

Author’s answer: Modification was done

R1#19: “Suggestion: I would use "These" instead of "Maternal effects".”

Author’s answer: Modification was done

R1#20: “Suggestion: I would use "in comparison to" instead of "to".”

Author’s answer: Modification was done

R1#21: “Are all these (lines 148-151) values refering to SD? if so, please include it.”

Author’s answer: Values correspond indeed to standard deviation, we specified it line 160.

R1#22: “Plastic plates or petri dishes? Please confirm. 25cm2 is the bottom area, right? if so, please include it.”

Author’s answer: Eggs were spread on plastic plates as mentioned in the text, not petri dishes. 25cm2 is indeed the bottom area. For clarity, we replaced the surface by length and height measurements (l. 168)

R1#23: “Please include model/brand and amplification.”

Author’s answer: Information were added to the text.

R1#24: “To what contaminants are the authors referring to? Some contaminants are not removable just with seawater. Moreover, the presence of some contaminants could have had a negative impact on embryo development and subsequent development stages.”

Author’s answer: Contaminants is probably not the most accurate term, we referred here to potential sperm/blood leftovers. Since it can be misinterpreted, we removed that term. Contaminants such as pollutants are mentioned in the discussion as a potential additional stressor (l. 634 and 747).

R1#25: “Which endpoint was used to assess if embryos were dead, upon MS-222 overdose exposure?”

Author’s answer: Herring embryos were considered dead when we could not observe any heartbeat. This was added to the text l.191, as well as the concentration used.

R1#26: “Suggestion: I would use "aquatic rearing systems" instead of just "rearing systems". there are other information (rearing systems) missing, namely: i) Were the rearing systems open, closed (RAS) or semi-open?; ii) Was there biological filtration?; iii) protein skimmers?”

Author’s answer: The text was modified to add “aquatic”. The system was semi-open (partial reuse), this information was added to the text. There was no specific biological filtration. Ammonia was controlled by water exchange. Oxygen was controlled with aeration. There was no need for protein skimmers as water was mechanically filtered down to 1 µm.

R1#27: “Suggestion: maybe it is better to change from "... UV-treated ... and mechanically filtered ..." to "... mechanically filtered ... and UV-treated". UV sterilization is always the last step of water treatment.”

Author’s answer: You are right, we modified the sentence.

R1#28: “The seawater was natural one, right? If so, please include it as natural seawater (NSW).”

Author’s answer: We modified it.

R1#29: “Suggestion: I would use "12h/12h L:D (light/dark cycle)" instead of just "12:12".”

Author’s answer: Modification was done.

R1#30: “What do the authors mean by "luminosity"? Maybe "illuminance" instead. Also lx, I do not understand the meaning of it. Maybe "LUX" instead. Please clarify.”

Author’s answer: Luminosity corresponds to light intensity, we modified the text. Lx corresponds to lux, we modified it.

R1#31: “Please include the measuring devices (model, brand).”

Author’s answer: We added the information to the text.

R1#31: “Please clarify because I do not understand the meaning of it, i.e. all larvae ... died in the egg??”

Author’s answer: We modified the text since it was not clear. What we meant is that the experiment ended when all larvae had hatched or when there was no more hatching (dead embryos). We were referring to embryos that died without hatching.

R1#32: “Suggestion: I would include (days post fertilization (dpf)" instead, both in the table caption and remaining text of the work”

Author’s answer: Since the table is independent from the text (journal guidelines), we introduced the abbreviation further in the text (line 273-274).

R1#33: “Could the authors provide the brand/model of the salinity measurement equipment that provides such accurate measurements? Moreover, are these salinity values representative of the collection areas, at the time of adult collection/trawling? Please clarify.”

Author’s answer: The brand of the multiparameter probe was added, as required earlier, line 209-210. These salinities are lower than the fishing zone since natural seawater is pumped close to the shore. However, salinity was shown to have no impact on fertilization rate above 20 according to Holliday & Blaxter (doi: 10.1017/S0025315400013564). According to the same paper, it can however impact morphological characteristics (egg diameter, length at hatching) and hatching rate. This parameter is therefore mentioned at the end of discussion as a driver to consider for future studies based on multi-driver approaches (l. 746).

R1#34: “Something is missing”

Author’s answer: Indeed, the small “i” (for total length at age i) disappeared during submission (word to pdf), we inserted it again.

R1#35: “Could the authors please clarify??”

Author’s answer: Since two out of the four plates were manipulated along development, there were excluded from hatching rate assessment to avoid biases. That is why hatching rate is calculated for two plates per family and per temperature regime (one hatching rate value per plate, therefore two values per family and per temperature scenario). Instead of “per female” it should be “per family”. Text and figures were modified since using “female” could be confusing. Overall we had: 2 plates x 4 families x 3 temperatures = 24 plates evaluated in total. One plate had a null fertilization rate and we thus we had a total of 23 plates. We modified the text to make it clearer.

R1#36: “YY axis description according to Figure 1caption”

Author’s answer: The figure axes were modified to match with the caption.

R1#37: “Please add afterwards "between temperature treatments."

Author’s answer: Text was modified.

R1#38: “Please provide reference(s)”

Author’s answer: This statement is not based on a specific article but results from what we can see directly on Figure 2. Figure 2 shows our estimates relatively to other Clupeidae. Data was extracted from Fishbase as mentioned in the material and methods l. 251. 

R1#39: Suggestion: Please include "However" before "these"

Author’s answer: Text was modified.

R1#40: “Besides the reference, please indicate which ones, i.e. some examples”

Author’s answer: More specific examples were provided.

R1#41: “Please include link: https://archimer.ifremer.fr/doc/00657/76942/78148.pdf”

Author’s answer: Link was added

R1#42: please include link

Author’s answer: Link was added

R1#43: please include link:https://www.ices.dk/sites/pub/CM%20Doccuments/1983/L/1983_L33.pdf

Author’s answer: Link was added

R1#44: please include link: https://www.biodiversityjournal.com/pdf/9(1)_19-24.pdf

Author’s answer: Link was added

Reviewer 2:

R2#1: “Reviewer #2: The manuscript entitled “Impact of temperature on Downs herring embryonic stages: first insights from an experimental approach” is a well-written record of the impact of temperature on Downs herring embryonic stages in a practical way. The MS explains a scientifically planned research work, and the results were expressed with sufficient data. The research is a piece of novel information for the scientific community to understand the dynamics of the impact of temperature on Downs herring embryonic stages, which is much more relevant today. Even though the MS is written well, there are a few suggestions to improve MS. With below mentioned minor corrections; the MS may be considered for publication in the journal.”

Author’s answer: We thank you for your evaluation, we made all modifications required in the title/text and we respond to all your specific points below.

R2#2: “Title: The scientific name can be included in the title.”

Author’s answer: Following your recommendation, we added the scientific name to the title and therefore removed the scientific name from key words and replaced it by “Atlantic herring”.

R2#3: “Abstract: Some of the values/statistics can be used in the abstract to explain the results of the present study to increase clarity. A sentence can be added at the end to demonstrate the relevance of the results or as a conclusion with the outcome of the study.”

Author’s answer: Following your recommendation, we added some information in the abstract to results, as well as a conclusion.

R2#4: Introduction: I felt the introduction is very elaborate and needs to be concise. It can be started with the background information, research problem, present status of the research in that line, and what the present study aims to unravel.

Author’s answer: We thank the reviewer for their suggestions, but do not feel it appropriate at the stage of “Minor revisions” to rewrite the entire introduction as the text is inherently both sequential and relevant to the background and questions of the paper. Nevertheless, we have shortened the text throughout the introduction to be more concise.

R2#5: “Materials and Methods: Properly written with all required details. Results and Discussion: Written well with good data representations.”

Author’s answer: We thank you for your evaluation.

Reviewer 3: 

R3#1: “Reviewer #3: This article aims to evaluate the effect of different water temperatures on the sensitive embryonic period and investigate potential parental effects on embryonic traits. The Ms in general presents well written and organized. For this reason, I think that after a minor revision this paper has to be published soon. Some comments are listed below”

Author’s answer: We thank you for your review and respond to all your points below.

R3#2: “Line 175-176: All animal experiments should comply with the ARRIVE guidelines. It seems that all breeders were killed for GSI and muscle RNA:DNA studies, please revise it.”

Author’s answer: Breeders were not killed since we got fish from fishermen and fish were already dead when reaching harbor. This specific point is raised in the discussion. We added it to material and methods line 159 to be clearer.

R3#3: “Line 223: Total Length, please use in lower case.”

Author’s answer: Modification was done.

R3#4: “Line 282: Same as above. Sac Volume, please use in lower case.”

Author’s answer: Modification was done.

R3#5: “Table 1: The authors should add explanations about experimental days under each temperature regime. To my knowledge, usually incubation time is given at half the day of yours (Blaxter, 1968).”

Author’s answer: Incubation times at the different temperatures are provided in Figure 2. For instance, it is in average 32 half-days at 8°C, which corresponds to 16 days. This is congruent with Blaxter (1968) and Blaxter & Hempel (1963, 1966). Days in Table 1 do not correspond to incubation times but to experimental duration considering batches 1 and 2. It takes therefore into account incubation time of batch 1 (which started 8th of December 2021) + incubation time of batch 2 (which started the 17th of December 2021), knowing that the two batches were overlapping. This seemed to be confusing so we removed it. Incubation time is now only represented in figure 1.

R3#6: “2.5. Statistical analyses: Statistical measures such as SD or SE should be identified..”

Author’s answer: This point was raised by Reviewer 1 (l.160) and we specified that we used SD. In all tables were variations are indicated, SD is indicated.

---

## [Editor Report · Decision Letter 1]

27 Mar 2023

Impact of temperature on Downs herring (Clupea harengus) embryonic stages: first insights from an experimental approach

PONE-D-22-34768R1

Dear Dr. Lola,

We’re pleased to inform you that your manuscript has been judged scientifically suitable for publication and will be formally accepted for publication once it meets all outstanding technical requirements.

Kind regards,

A. K. Shakur Ahammad, PhD

Academic Editor

PLOS ONE
---

## [Editor Report · Acceptance letter]

30 Mar 2023

PONE-D-22-34768R1 

Impact of temperature on Downs herring (*Clupea harengus*) embryonic stages: first insights from an experimental approach 

Dear Dr. Toomey:

I'm pleased to inform you that your manuscript has been deemed suitable for publication in PLOS ONE. Congratulations! Your manuscript is now with our production department. 

Kind regards, 

on behalf of

Dr. A. K. Shakur Ahammad 

Academic Editor

PLOS ONE